# Traumatic Intralenticular Neovascularization in a HLA B27+ Pediatric Patient

**DOI:** 10.3390/diagnostics11081493

**Published:** 2021-08-18

**Authors:** Călin Petru Tătaru, Cătălina Ioana Tătaru, Maria Dudău, Alexandra Moșu, Lăcrămioara Luca, Bosa Maria, Alice Bancu, Paul Filip Curcă

**Affiliations:** 1Clinical Department of Ophthalmology, Carol Davila University of Medicine and Pharmacy, 020021 Bucharest, Romania; calinpetrutataru@yahoo.com (C.P.T.); filipcurca@yahoo.com (P.F.C.); 2Department Ophthalmology I, Clinical Hospital for Ophthalmological Emergencies, 010464 Bucharest, Romania; dudau_maria2002@yahoo.com (M.D.); alexandra.mosu@hotmail.com (A.M.); 3Department of Cellular and Molecular Biology and Histology, Carol Davila University of Medicine and Pharmacy, 020021 Bucharest, Romania; 4Anatomopathological Laboratory of the National Institute for Legal Medicine “Mina Minovici”, 042122 Bucharest, Romania; lacra.luca@gmail.com (L.L.); maria.bosa@yahoo.com (B.M.); alice.bancu@yahoo.com (A.B.)

**Keywords:** lens tumor, masquerade tumor, lens vascularization, neovascularization, HLA B27, CD68, foamy macrophages

## Abstract

(1) Background: Intralenticular tumors are an entity akin to Schrodinger’s cat since, although the human crystalline cells themselves are not known to malignly proliferate, various entities can take the appearance and clinical presentation of a tumor originating in the lens. We present the peculiar case of an 11-year-old male patient of African descent, HLA B27+, with a previous history of minor ocular trauma and unilateral anterior uveitis a year before which was admitted to our department with total opacification of the crystalline lens in the right eye and lens neovascularization. During surgery, a vascular, white fibrotic mass measuring 0.1–0.2 cm was discovered inside the lens bag and was excised. (2) Methods: Retrospective case review. (3) Results: The histopathological exam of the excised mass revealed an abundant infiltrate consisting of CD68+ foamy macrophages and lymphoplasmacytic elements. CD68 is a pan-macrophage marker associated with an active inflammatory mechanism soliciting macrophages, and tissue activated macrophages are correlated to increased stromal and serum levels of vascular endothelial growth factor, providing an explanation for lens angiogenesis. (4) Conclusions: The diagnosis is of a “masquerade tumor” resulted from an abnormal inflammatory process in connection with previous ocular trauma and possibly the patient’s HLA B27+ status.

## 1. Introduction

Tumors concerning the lens (intralenticular) have been clinically described, however, the human crystalline cells are atypical since, according to Albert DM. et al.’s [1] study of 18,000 patients and 45,000 ocular veterinary cases, no tumors with lens origin could be found in humans which is in stark contrast to other mammals such as domestic cats [2]. Veromman [3] concluded that although no evidence of malignant tumoral growth was found in human crystalline cells, abnormal and benign growths of lens cells can present with cell atypia [3] and associate with cataract formation. While seemingly paradoxically alike with Schrodinger’s cat [4], inflammatory mases related to previous episodes of uveitis [5] or melanoma metastases [6] were documented to disguise as lens masses under the term “Masquerade tumors”.

The adult human lens is an avascular tissue under normal circumstances. The formation of the lens originates in a subset of placodal progenitor cells situated between the anterior neural plate and surface ectoderm that, having expressed the Pax6 transcription factor, differentiate into lens cells as opposed to olfactory and adenohypophysis cells [7]. This process requires adequate blood supply through the embryonic hyaloid vasculature network, which normally regresses after the complete formation of the lens [8]. Persistent fetal vasculature (PFV) is the congenital anomaly where this regression is incomplete [8], first being described by Cloquet [9]. Concerning that the lens PFV presents as persistent tunica vasculosa lentis (TVL), the failure of regression of the anterior TVL results in abnormal iridohyaloid blood vessels and failed regression of the posterior TVL results in the formation of a retrolental membrane [8], which can present various sized from barely noticeable to sprawling, dense fibrosis. After the complete formation of the lens and regression of fetal vasculature, a balance is reached where the conundrum of antiangiogenetic and angiogenetic factors maintain an avascular lens tissue [10,11]. This balance is perturbed by chronic hypoxemia that flips the balance in favor of angiogenesis resulting in neovascularization of the posterior lens capsule in cases of retinal detachment or proliferative diabetic retinopathy in pseudophakic eyes [12] and, furthermore, in rare cases of lens trauma and traumatic cataract formation [10,13,14].

## 2. Materials and Methods

Retrospective case review including a search of our hospital records for previous presentations of the patient.

## 3. Results

### 3.1. Case Presentation

An 11-year-old male patient of Central African descent, HLA B27+ positive and currently under treatment with methotrexate, was admitted to our department with progressively decreased visual acuity and leukocoria in the right eye.

Anamnesis revealed a minor traumatic episode (fingernail scratch) a year before the current presentation involving the right eye. A detailed search of hospital records produced a previous presentation 3 months after the initial traumatism, where the patient was diagnosed with right eye anterior uveitis with hypopyon and posterior subcapsular cataract. The patient underwent local mydriatic and local and systemic anti-inflammatory and antibiotic treatment with a complete remission of the uveitis, and the patient was subsequently referred to rheumatology and infectious diseases specialists and was lost to follow-up until the current presentation.

A complete ophthalmological exam and slit-lamp biomicroscopy revealed a completely opaque lens in the right eye (Figure 1A) with fine blood vessels apparently located inside the lens (Figure 1B). A presumptive diagnosis of complicated cataract subsequent to previous uveitis was established. Right eye B mode ultrasound revealed an attached retina and no vitreous abnormalities. Biometry was performed and resulted in a similar axial length of both eyes (23.60 mm) and a difference in cylindrical diopters, with −1.09 D at 22° for the right eye versus the left eye’s −0.7 D at 148°. A 21 diopter, three-piece intraocular lens (IOL) was chosen.

Surgery was planned and performed. During the surgery (Figure 2), the anterior capsule was firmly attached to the lens by fibrous adherences (Figure 2B–D) which presented neovascularization, and after a small amount of blood was discharged manipulation (Figure 2C). After the aspiration of the lens material, a vascular fibrotic tissue previously covered by the cortex and anterior capsule was discovered in the lens bag (Figure 2E), with a greater consistency than the lens and a visible blood-vessel traversing its length. The specimen was excised (Figure 2F) and sent for histopathological examination. A fibrous vascularized proliferation remains attached in the supero-temporal region of the posterior capsule (Figure 2G). A posterior capsulorhexis (Figure 2H) with limited anterior vitrectomy and the excision of the fibrous proliferation were performed. The lens was implanted using the technique of posterior optic capture, and the surgery was conducted without other complications (Figure 2I). Patient evolution was uneventful and was discharged after 2 days. An additional movie file presents a video synopsis of the surgery with the previously mentioned surgical steps (see additional file Appendix A Video synopsis of the surgery).

The patient presented for the latest follow-up examination a year after the surgery and 10 months after cessation of the immunosuppressive treatment. Uncorrected visual acuity (UCVA) was 20/20 (1.0 decimal, log MAR 0) with a refraction of −0.5 spheric diopters and −0.25 cylinder at 23°. No signs of inflammation or neovascularization were observed, and no tumoral reoccurrence was observed.

### 3.2. Histopathological Examination of the Specimen

The macroscopic aspect is of a white, consistent tumor about 0.1–0.2 cm in length.

Histopathological analysis of the excised specimen was performed using standard Hematoxylin and eosin stain (HE) and Van Gieson staining along with immunohistochemistry.

A fibro-vascular tissue fragment was identified with resection limits, containing a stroma rich in immature and loose collagen fibers, stained with Van Gieson (Figure 3). An abundant infiltrate consisting of foamy macrophages, evidenced by using CD68 staining antibodies, and lymphoplasmacytic elements was observed. The tissue did not contain any foreign body. Figure 3, Figure 4, Figure 5, Figure 6 and Figure 7 present in detail these elements.

## 4. Discussion

Our case presents the diagnostic difficulties of finding an intralenticular vascularized mass inside an opacified lens and, furthermore of pinpointing its etiology. The presumptive diagnosis after clinical examination was complicated cataract due to previous uveitis episodes. In light of the intraoperative findings and the subsequent histopathological analysis of the specimen, the following additional differential diagnoses were considered:

A retrolental membrane, a congenital development abnormality with origin in the persistence of fetal vasculature (PFV) [8], with a posterior tunica vasculosa lentis (TVL) resulting in the formation of a retrolental membrane can present various sized from barely noticeable to sprawling, dense fibrosis. This membrane could also rupture during a traumatic event, resulting in the formation of a secondary cataract considering the patient recounted a traumatic incident. However, our 11-year-old patient did not have amblyopia associated with such a congenital abnormality. This diagnosis was also excluded based on the patient’s previous presentation for anterior uveitis, where neovascularization was absent.

Neoplastic syndromes are known to masquerade as uveitis and characteristically present as reoccurring unilateral uveitis with poor response to treatment [15]. These include multiple types of lymphoma (such as primary intraocular/vitreoretinal lymphoma, mucosal associated lymphoid tissue lymphoma of the optic nerve sheath and Non-Hodgkin’s Lymphoma), leukemia (such as chronic lymphocytic leukemia and human T-lymphotropic virus T cell leukemia) and metastases of melanoma [15]. The histopathological examination of the specimen, young age of the patient (11 years old) and anterior localization resulted in the exclusion of these diagnoses.

The immunohistochemistry exam of the specimen highlighted the presence of CD68+ foamy macrophages. Microsialine (CD68) is a D-class scavenger receptor that consists of a glycosylated type I membrane protein from the lysosome-associated membrane protein family [16,17]. Several cells express this membrane protein, including osteoclasts and especially macrophages. CD68 is used as a pan-macrophage marker [17], and its intense expression is associated with highly-active inflammatory mechanisms soliciting macrophages. CD68+ foamy macrophages are associated with immune-mediated inflammation, appearing in reactive vitreous infiltrates in uveitis [18], phacolytic glaucoma [19], intraocular melanoma [20] or inflammatory processes involving other eye structures such as the iris in cases of antinuclear-antibodies-positive (ANA+) anterior uveitis associated with oligoarticular Juvenile Idiopathic Arthritis (JIA) [21]. The high density of tissue activated macrophages (TAM) was found to correlate with increased stromal and serum levels of vascular endothelial growth factor (VEGF) [17]. The chemotaxis of the CD68+ foamy macrophages to the lens material is possibly associated with interferon-gamma (IFNγ) [22]. This would imply IFNγ-related M1 macrophage polarization [23,24] or a complement activation model via C3 derived products [22]; however, M2 macrophage polarization could also be a possibility due to this pathway’s association with reparative tissular processes but would be abnormal, such as in the present case.

HLA B27 associations with Ankylosing Spondylitis (AS) and other related pathologies are lower in African populations compared to European populations with the prevalence of HLA alleles being exceedingly rare (<1%) [25], especially in Central African populations [26] similar to our patient. However, compared to Caucasian children, uveitis in African-American children is markedly associated with a more severe visual prognosis [27] and bilateral eye disease, with more frequent complications such as development of cataracts, glaucoma, synechiae, band keratopathy and cystoid macular edema and also an increased rate of development of unique complications [27]. Poorer outcomes were especially observed in African-American children with non-JIA associated uveitis [27]. In the case of clinically manifest Ankylosing Spondylitis (AS) associated with HLA B27+ status, African populations have higher overall disease activity and progression with more comorbidities [28,29] and poorer visual outcomes in case of uveitis [30] and were more likely to present with hypopyon [30]. There are variations depending on the African region [26], with certain population with high HLA B27 carrier rates and absence of spondyloarthropathies; only HLA-B * 14:03 and HLA-B * 27:05 alleles were significantly increased in Central African AS patients [26]. As such, the HLA B27+ status in our patient could have contributed to the overall inflammatory process and be linked to the patient’s previous presentation for anterior uveitis. The greater association of unique complications with uveitis in African-American patients is noteworthy to our case [27].

All patients with multiple associated ocular complications undergoing cataract surgery should benefit from a thorough preoperative examination (including anamnesis, clinical, ocular and family history, laboratory tests, slit-lamp biomicroscopy, ocular ultrasound and ultrasound biomicroscopy/optical coherence tomography), keeping in mind that inflammation and trauma should be approached carefully. Although our case was unique, the therapeutic approach was standard: Initially, we treated the inflammation and investigated its possible etiologies, and we performed cataract surgery once the lens opacification impaired the patient’s vision. The intralenticular mass was an intraoperative finding; therefore, it was handled with care and sent for further investigation. Prior to surgery, the patient underwent treatment for chronic idiopathic uveitis. During the postoperative period, the medical therapy for the uveitis was ceased and no recurrence was noted.

## 5. Conclusions

The most likely complete diagnosis is a “Masquerade tumor”, an inflammatory mass related to the previous minor trauma and possibly linked to the patient’s HLA B27+ status. The intraoperatively intact anterior lens-capsule does not exclude a traumatic origin of the process as there is the possibility of tissular repair of the breach from the moment of the trauma to the present moment of the patient’s examination, which is a whole year later. Considering the histopathological findings, the mass corresponds to an exaggerated inflammatory response possibly linked to the patient’s HLA B27+ status with the recruitment of inflammatory elements, including high amounts of CD68+ foamy macrophages and the local production of VEGF. We hypothesize that this inflammatory cascade repeated itself in a vicious circle where inflammatory elements promoted vascular growth that, in turn, brought more inflammatory elements to the scene, resulting in an abnormal tissue regeneration resembling an opaque, cataracted lens.

### 5.1. What Was Known

Human lens cells are not known to malignly proliferate.

“Masquerade tumors” are tumors that can mimic the appearance and clinical presentation of a tumor originating in the lens. Lens neovascularization can be present.

### 5.2. What This Paper Adds

Our case presents such a “masquerade tumor” in the context of a history of initial minor lens trauma and subsequent uveitis associating HLA B27+ status.

Lens trauma can jump-start immune-mediated inflammatory ocular processes with the generation of neovascularization and abnormal granulation tissue.

## Figures and Tables

**Figure 1 diagnostics-11-01493-f001:**
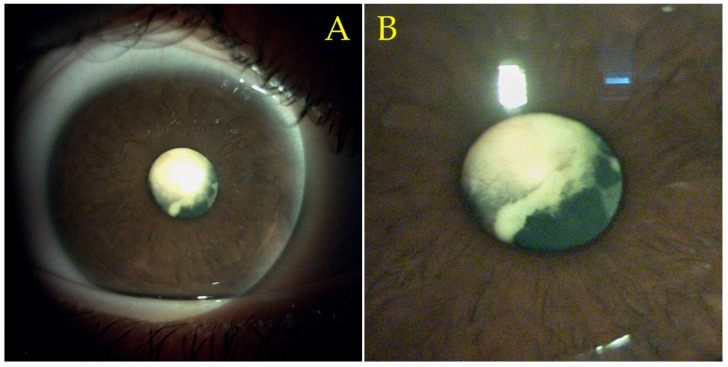
16x magnification slit-lamp biomicroscopy images of the patient’s right eye. (**A**) Reveals an opaque lens. (**B**) Enhanced (zoomed-in) close-up image that reveals the presence of fine blood vessels located intralenticularly.

**Figure 2 diagnostics-11-01493-f002:**
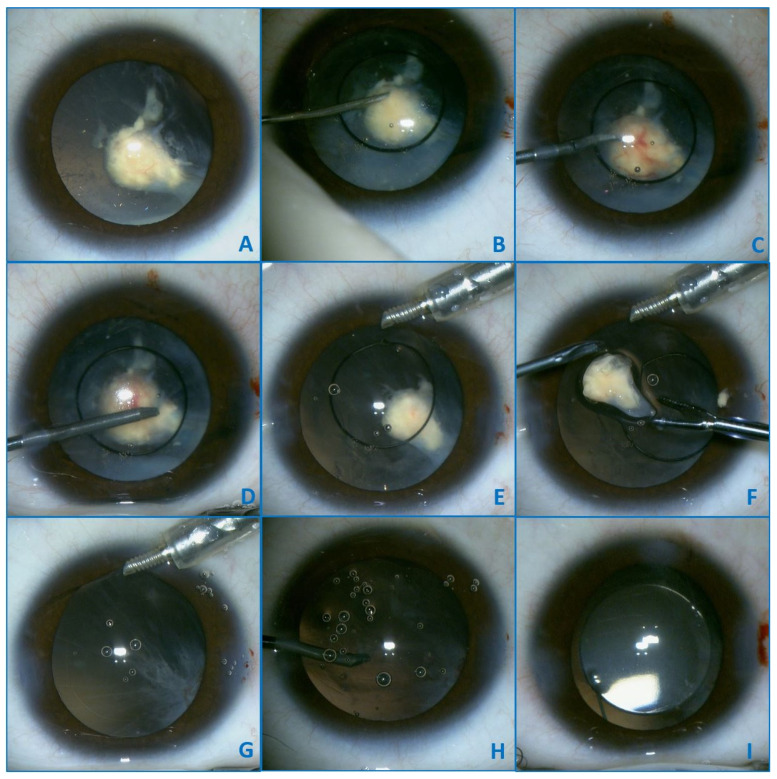
Intraoperative images—Initial aspect (**A**). During the surgery, the anterior capsule was firmly attached to the lens by fibrous adherences (**B**–**D**), which presented neovascularization, and a small amount of blood was discharged after manipulation (**C**). After aspiration of the lens material a vascular fibrotic tissue previously covered by the cortex and anterior capsule was discovered in the lens bag (**E**), with a greater consistency than the lens and a visible blood-vessel. The specimen was excised (**F**) and sent for histopathological examination. A fibrous vascularized proliferation remains attached in the supero-temporal region of the posterior capsule (**G**). A posterior capsulorhexis (**H**) with limited anterior vitrectomy and the excision of the fibrous proliferation were performed. The lens was implanted by using the technique of posterior optic capture, and the surgery was conducted without other complications (**I**).

**Figure 3 diagnostics-11-01493-f003:**
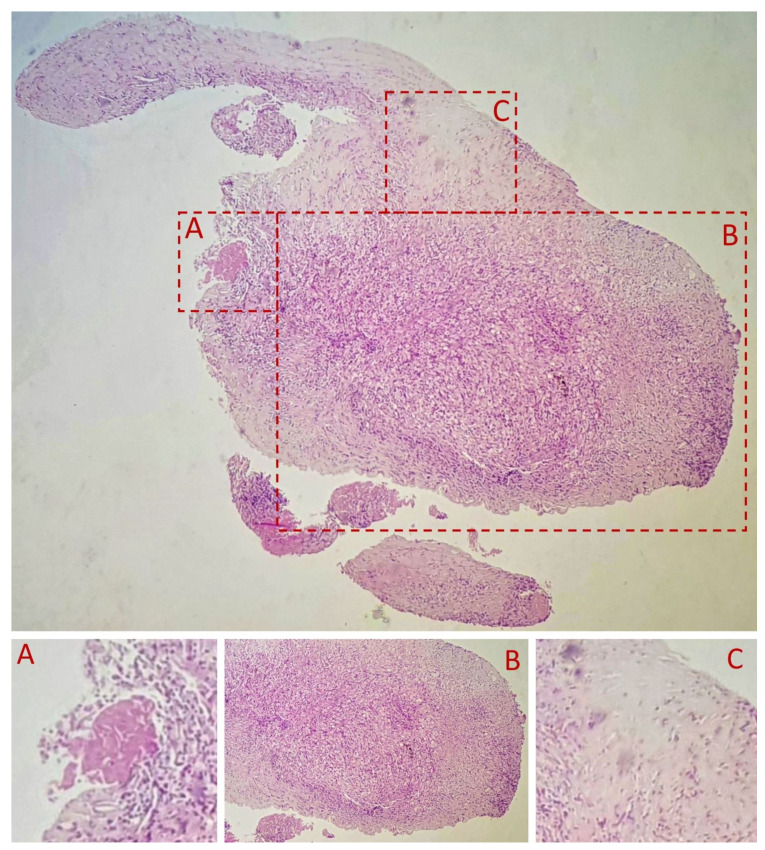
HE stain, 4×. (**A**) indicates a resection artifact. (**B**) The inflammatory infiltrate of foamy macrophages and lymphoplasmacytic elements. (**C**) indicates an abundance of immature collagen fibers.

**Figure 4 diagnostics-11-01493-f004:**
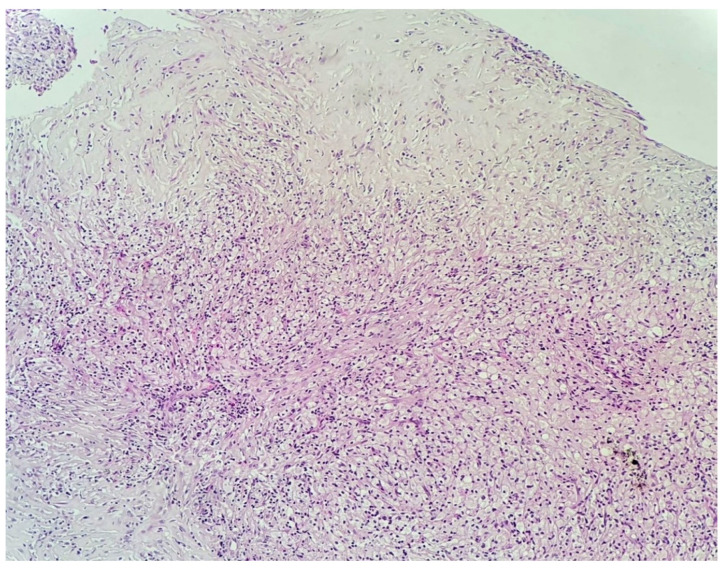
HE stain, 10× lens. Foamy macrophages predominantly centrally located. Numerous loose collagen fibers are present throughout the specimen, with more located toward the periphery.

**Figure 5 diagnostics-11-01493-f005:**
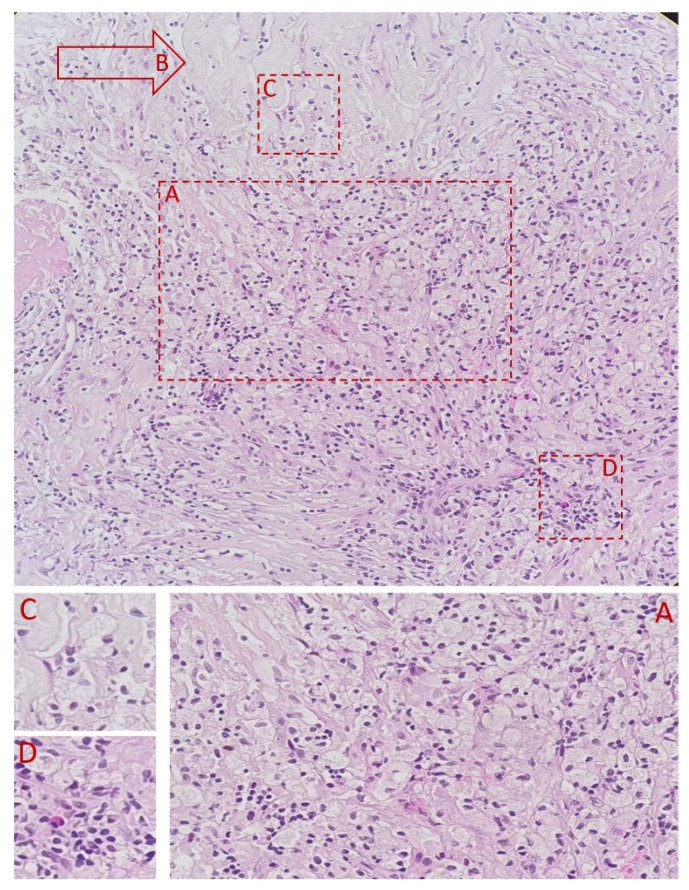
HE stain 40× lens—close-up view of the inflammatory infiltrate. (**A**) highlights the foamy macrophages. (**B**) indicates immature collagen fibers. (**C**,**D**) note the presence of lymphoplasmacytes.

**Figure 6 diagnostics-11-01493-f006:**
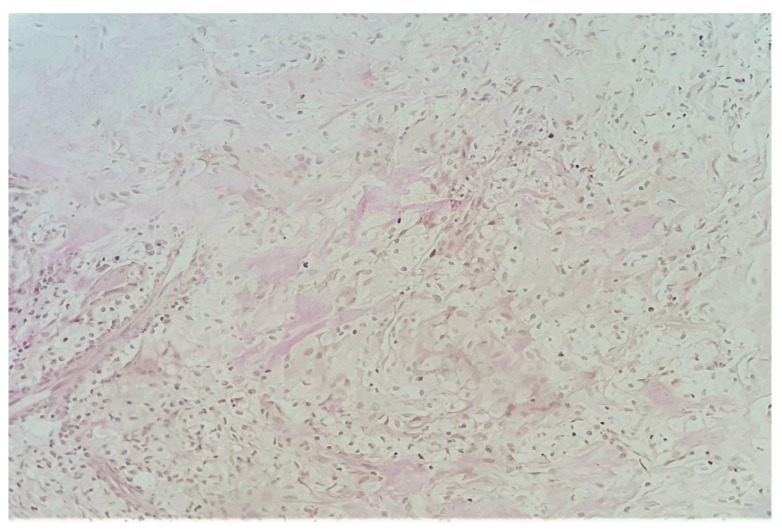
Van Gieson’s stain, 20× lens—close-up of the immature collagen fibers.

**Figure 7 diagnostics-11-01493-f007:**
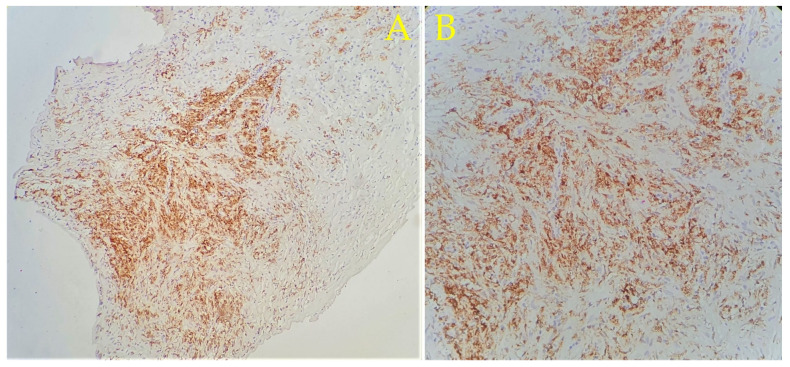
Immunohistochemical analysis using CD68 staining antibodies. (**A**) using 10× lens: the overall picture is of a highly metabolically active granulation tissue that is not yet matured. (**B**) using 20× lens: intense positivity and diffuse capture in the cytoplasm of the macrophages, identifying them as CD68+ foamy macrophages.

## Data Availability

Data and materials regarding the clinical presentation of the patient, surgery and post-operative evolution pertain to our ophthalmological department. Histopathological data pertains to the Anatomopathological laboratory of the National Institute for Legal Medicine “Mina Minovici” Bucharest.

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
