# Peer review of "Traumatic Intralenticular Neovascularization in a HLA B27+ Pediatric Patient"

_diagnostics, 2021, doi:10.3390/diagnostics11081493_

Round 1
Reviewer 1 Report
This is a well thought and interesting retrospective case study review focused on the diagnosis and surgical treatment of “masquerade tumor” related to an abnormal inflammatory process due previous ocular trauma and possibly the HLA B27+ status of the 11 year-old male patient of African descent. Both the surgical treatment and the histopathological exam of the excised mass are well described and documented so that those to be of direct help of the clinical community. In the context of intralenticular tumors studies it is traditionally well known that human lens cells in general do not malignly proliferate and that “masquerade tumors” are observed in which case neovascularization can be present. The novelty of this case study is that a “masquerade tumor” is presented in the context of a history of minor lens trauma and subsequent uveitis and HLA B27+ status. Thus minor lens trauma is shown to be able to initiate immune-mediated inflammatory ocular processes with occurrence of neovascularization and abnormal granulation tissue.
My only minor comment is to slightly extend the discussion section in direction of explanation of the expected frequency of such type of pathological complications in different racial types and also to summarize in one sentence what is the clinical algorithm of diagnostic and treatment steps recommended by the authors when in doubt for such complication into the patient.
Author Response
Dear Sir/Madam,
Thank you for your useful suggestions and comments.
The requested information has been added to the manuscript.
Regarding the frequency of such complications in different racial types we extended the discussion section by adding information from the literature:
"Compared to Caucasian children, uveitis in African-American children is markedly associated with a more severe visual prognosis and bilateral eye disease, with more frequent complications such as development of cataracts, glaucoma, synechiae, band keratopathy and cystoid macular edema and also an increased rate of development of unique complications. Poorer outcomes were especially observed in African-American children with non-JIA associated uveitis.... The greater association of unique complications with uveitis in African-American patients is noteworthy."
Although we work in a very big hospital with emergency unit such cases as presented in the manuscript were extremely rare encountered.
At the discussion section we added as well our diagnostic and therapeutic approach of such cases which are represented by standard rigorous and detailed patient examinations and careful medical and surgical treatment.
"All patients with multiple associated ocular complications undergoing cataract surgery should benefit of a thorough preoperative examination (including anamnesis, clinical, ocular and family history, laboratory tests, slit-lamp biomicroscopy, ocular ultrasound and ultrasound biomicroscopy /optical coherence tomography) always considering that inflammation and trauma should be approached carefully. Although our case was unique, the therapeutic approach was standard: initially we treated the inflammation and investigated its possible etiologies and once the lens opacification impaired the patient’s vision, we performed cataract surgery. The intralenticular mass was an intraoperative finding and therefore it was handled with care and sent for further investigations. Prior to surgery the patient underwent treatment for chronic idiopathic uveitis. During the postoperative period the medical therapy for the uveitis was ceased and no recurrence was noted."
Thank you for your time and we hope our modifications will meet your expectations.
Corresponding author,
Cătălina Ioana Tătaru MD PhD
Reviewer 2 Report
The authors present an extremely rare and fascinating case. The step by step approach of the case (initial thought of complicated cataract that turned out to be a surprise in surgery) alerts the reader regarding the management of such cases and the precautions needed when operating on such patients.
In an otherwise high quality presentation, the only part that could be a little different is the similarity in the first sentence of the abstract and introduction (ie rephrase maybe the introduction sentence).
Author Response
Dear Sir/Madam,
Thank you for your useful suggestion.
We adress the issue regarding the introduction and the repeated phrase in the abstract as following:
Abstract:
“Intralenticular tumors are an entity akin to Schrodinger’s cat as although the human crystalline cells themselves are not known to malignly proliferate, various entities can take the appearance and clinical presentation of a tumor originating in the lens.”
Introduction:
“Tumors concerning the lens (intralenticular) have been clinically described, however the human crystalline cells are atypical since, according to Daniel M. Et. All [1] study of 18000 patients and 45000 ocular veterinary cases, no tumors with lens origin could be found in humans, in stark contrast to other mammals such as domestic cats [2].”
Thank you for your time and we hope the manuscript will meet your expectations.
Corresponding author,
Cătălina Ioana Tătaru MD PhD